# Improved Weighted Tensor Schatten $p$-Norm for Fast Multi-view Graph Clustering

Yinghui Sun
Southeast University
Nanjing, China
sunyh@seu.edu.cn

Xingfeng Li*
Nanjing University of Science and Technology
Nanjing, China
lixingfeng@njust.edu.cn

Quansen Sun
Nanjing University of Science and Technology
Nanjing, China
sunquansen@njust.edu.cn

Min-Ling Zhang
Southeast University
Nanjing, China
zhangml@seu.edu.cn

Zhenwen Ren
SongShan Laboratory
Henan, China
rzw@njust.edu.cn

## ABSTRACT

Recently, tensor Schatten $p$-norm has achieved impressive performance for fast multi-view clustering [57]. This primarily ascribes the superiority of tensor Schatten $p$-norm in exploring high-order structure information among views. Whereas, 1) tensor Schatten $p$-norm treats different singular values equally, such that the larger singular values corresponding to certain significant feature information (i.e., prior information) have not been utilized fully; 2) tensor Schatten $p$-norm also ignore ranking the core entries of core tensor, which may contain noise information; 3) existing methods select fixed anchors or averagely update anchors to construct the neighbor bipartite graphs, greatly limiting the flexibility and expression of anchors. To break these limitations, we propose a novel **Improved Weighted Tensor Schatten $p$-Norm for Fast Multi-view Graph Clustering (IWTSN-FMGC)**. Specifically, to eliminate the interference of the first two limitations, we propose an improved weighted tensor Schatten $p$-norm to dynamically rank core tensor and automatically shrink singular values. To this end, improved weighted tensor Schatten $p$-norm has the potential to more effectively leverage low-rank structures and prior information, thereby enhancing robustness compared to current tensor Schatten $p$-norm methods. Further, the designed adaptive neighbor bipartite graph learning can more flexibly and expressively encode the local manifold structure information than existing anchor selection and averaged anchor updating. Extensive experiments validate our effectiveness and superiority across multiple benchmark datasets.

## CCS CONCEPTS

• **Theory of computation → Theory and algorithms for application domains**; **Unsupervised learning and clustering**.

*Corresponding author.

## KEYWORDS

Fast multi-view clustering, adaptive neighbor bipartite graph learning, improved weighted tensor Schatten $p$-norm

**ACM Reference Format:**
Yinghui Sun, Xingfeng Li, Quansen Sun, Min-Ling Zhang, and Zhenwen Ren. 2024. Improved Weighted Tensor Schatten $p$-Norm for Fast Multi-view Graph Clustering. In *Proceedings of the 32nd ACM International Conference on Multimedia (MM '24), October 28-November 1, 2024, Melbourne, VIC, Australia Proceedings of the 32nd ACM International Conference on Multimedia (MM'24), October 28-November 1, 2024, Melbourne, Australia.* ACM, New York, NY, USA, 10 pages. https://doi.org/10.1145/3664647.3681334

## 1 INTRODUCTION

Recently, technological advancements have facilitated the collection and storage of extensive data from diverse sources or feature extractors, resulting in the emergence of multi-view data [4, 26, 27, 32, 36]. For instance, a single news story could exist in various formats like video, audio, or text. Meanwhile, each news could be also reported in different languages across nations such as Chinese, English, Russian, or French. Multi-view clustering (MVC) leverages the similarities between samples and views to effectively integrate the consistent and complementary attributes of unlabeled multi-view data, thereby categorizing them into relevant clusters [2, 3, 38, 45, 52, 55]. MVC has witnessed significant growth in recent times [1, 32, 35, 44, 48, 49].

Despite the myriad of MVC methods, scalability remains a paramount challenge in real-world large-scale applications [24, 25, 46, 61]. Current MVC methods often exhibit quadratic or cubic complexity with respect to the number of instances n, posing significant obstacles in handling extensive datasets [6, 7, 31, 33, 37]. To address this challenge, the anchor graph strategy has emerged as an effective solution [5, 21, 47, 62]. Initially, it selects m independent anchor points to comprehensively represent instances within each view. The quality of these anchors significantly impacts performance and is typically determined through random sampling, k-means, or heuristic sampling strategies (such as VDA anchor selection [57]). For instance, some studies leverage k-means operations and heuristic sampling on individual views, then integrate the outcomes into a consensus graph for final clustering results. However, these static anchor selection strategies limit the expressiveness and flexibility of anchors. In contrast, dynamic anchor learning methods aim to

learn and optimize anchors alongside model variations, enhancing both performance and adaptability.

Existing anchor learning strategies could be broadly divided into two categories [8, 18, 23]: The first one resembles the self-expressive property, which reconstructs each sample by combining learned anchors linearly from the original data [50, 51]. Then, the connections between samples and anchors produce a bipartite graph. Recent advancements in anchor-based multi-view clustering have stemmed from this strategy. Different from the first one, the second a small amount of neighbor bipartite graph learning methods to build the bipartite graph by assigning each sample some nearest anchors as neighbors, which could better capture the local manifold structure than the first one [19]. However, note herein that existing adaptive neighbor bipartite graph learning methods optimize anchors using average values across iterations, potentially leading to the loss of significant and flexible information. Adaptive neighbor bipartite graph learning is sensitive to noise or outliers since noise and outliers can destroy the local manifold structure in anchor-sample space. Moreover, these methods often overlook higher-order information that could enhance anchor quality. Due to the effectiveness of the tensor Schatten $p$-Norm in better approximating the target rank than the tensor nuclear norm, one may use tensor Schatten $p$-Norm to explore the high-order information of bipartite graphs [57]. However, existing tensor Schatten $p$-norm still suffer from the following limitations: (1) existing tensor Schatten $p$-norm treats singular values of tensor equally in each slice; (2) existing tensor Schatten $p$-norm cannot exploit the low-rank property fully in the core tensor space.

To address these issues, we proposed a novel fast local multi-view clustering method, termed Improved Weighted Tensor Schatten $p$-Norm for Fast Local Multi-view Clustering (IWTSN-FMGC). IWTSN-FMGC learns the bipartite graph by an adaptive neighbor bipartite graph learning, which usually better protects the local manifold structure than the widely-used subspace self-representation learning. And local manifold structure plays a significant role in clustering. Our adaptive neighbor bipartite graph learning adaptively updates the anchors with orthogonal constraints rather than averaging the value of anchors. Furthermore, our IWTSN-FMGC develops a new IWTSN to consider the prior information of different singular values. Then, IWTSN can also better exploit the low-rank property of core tensor by further ranking the diagonal core entries of core tensor. In this way, IWTSN of our IWTSN-FMGC could enhance the robustness of the adaptive neighbor bipartite graph learning to noise and outliers. Then, IWTSN-FMGC could learn more robust neighbor bipartite graphs to average as a final bipartite graph for obtaining the clustering results. In conclusion, this study makes the following contributions:

- We propose a novel Improved Weighted Tensor Schatten $p$-Norm (IWTSN) to better rank the core tensor and take advantage of the prior information from singular values, enhancing robustness and validity compared to the widely-used tensor Schatten $p$-norm [57]. Meanwhile, an elegant solver is developed to optimize the proposed IWTSN.
- By seamlessly coupling adaptive neighbor bipartite graph learning and IWTSN into a unified framework. Both procedures negotiate with each other to enhance the quality and

robustness of neighbor bipartite graphs. Thus, the more refined and robust neighbor bipartite graphs could be learned to better capture the local manifold structure information.
- Through extensive experiments, the proposed method showcases its superiority over state-of-the-art methods.

## 2 RELATED WORK

### 2.1 Neighbor Graph based MVC

Generally, adaptive neighbor graph learning learns a neighbor graph by assigning a probability value to a sample as the neighbor similarity of another one [39, 53]. In this way, the local manifold structures between samples could be captured, which facilitates the clustering partition [11]. Although effective, the time and space complexities are cubic and quadratic to sample number, respectively. Thus, these methods limit the application for large-scale tasks. Furthermore, many advances employ the tensor nuclear norm or tensor Schatten $p$-norm to further explore the high-order information among neighbor graphs [19, 56, 57], *i.e.*,

$$\min_{\mathcal{L}, \hat{A}} \sum_{i=1}^{n} \sum_{j=1}^{n} \mathbb{D}(\mathbf{x}_i^r, \mathbf{x}_j^r)\hat{a}_{ij}^r + \beta\|\mathcal{L}\|_{\circledast} \tag{1}$$

where the larger distance criterion $\mathbb{D}$ has the smaller similarity value $\hat{a}_{ij}$ of similarity graph $\hat{A} \in \mathbb{R}^{n \times n}$ [15] in the $r$-th view. Control parameter $\beta$ tunes the contribution between adaptive neighbor graph learning and tensor constraints, such as tensor nuclear norm $\|\mathcal{L}\|_{\circledast}$ or tensor Schatten $p$-norm $\|\mathcal{L}\|_{s_p}^p$ [19, 54, 56, 57]. Regrettably, these methods also suffer from time- and space- prohibitive. More importantly, these methods cannot explore the high-order low-rank property fully since they fail to take advantage of the prior information of singular values in the core tensor space.

### 2.2 Neighbor Bipartite Graph based MVC

To alleviate these limitations, neighbor bipartite graph construction is proposed. Neighbor bipartite graph $\mathbf{A}^r \in \mathbb{R}^{m \times n}$ of $r$-th view is constructed by linking their vertices by defining sample sets $\mathbf{X}^r \in \mathbb{R}^{d \times n}$ and anchor sets $\hat{\mathbf{B}}^r \in \mathbb{R}^{d \times m}$. Vertices represent the relationship between $m$ anchors and $n$ samples [28, 29, 62, 63] in $d$-dimension space. Anchors are typically strategically selected from the original space to approximate all samples. Different strategies are then employed to construct the bipartite graph for clustering. Due to its efficiency, various methodologies have made promising progress in recent years, with anchor learning strategies broadly categorized into subspace graph construction and neighbor graph construction. Unlike subspace graph construction, neighbor graph construction emphasizes preserving locality, assuming that the primary high-dimensional feature space exists within a lower-dimensional manifold. Consequently, neighbor graph construction often excels in preserving the low-dimensional manifold structure. Specifically, for the $i$-th sample, the $j$-th anchor is connected as a neighbor with a probability represented by $a_{ij}$. Intuitively, shorter distances between anchor-node pairs correspond to higher probabilities $a_{ij}$, and vice versa, as explained by

$$\min_{\mathbf{A}^r} \sum_{i=1}^{m} \sum_{j=1}^{n} \mathbb{D}(\hat{\mathbf{b}}_i^r, \mathbf{x}_j^r)a_{ij}^r \ \text{s.t.}(\mathbf{A}^r)^\top \mathbf{1} = \mathbf{1}, \mathbf{A}^r \geq 0 \tag{2}$$

With the different criteria $\mathbb{D}$, Eq. (2) captures the local manifold structures in different criterion spaces [19, 35, 55]. $\hat{\mathbf{b}}_i$ denotes the $i$-th anchor, and In Eq. (2), anchors are selected using k-means [13] or heuristic sampling methods [10, 22, 57]. These anchors remain fixed throughout the optimization process, known as static anchor selection, which limits their expressiveness and flexibility. Some researchers explore dynamic anchor learning, optimizing anchors alongside model variations. However, anchors learned from them are optimized using average values across iterations [1], still limiting the diversity and flexibility of anchors.

## 3 METHODOLOGY

### 3.1 Notations and Preliminaries

Notations and definitions of tensor and matrix Schatten $p$-norm and rank approximation [6, 57] are as follows.

Vectors and their components are denoted by bold lowercase letters ($\mathbf{k}$ and $k_{ij}$). Matrices are represented by bold uppercase letters ($\mathbf{K}$), while tensors are indicated by bold calligraphy letters ($\mathcal{K}$). For simplicity, we use the notation $\mathcal{K}(:,:,k)$ as $\mathbf{K}^k$ or $\mathcal{K}^k$. The fast Fourier transformation (FFT) and its inverse along the third dimension of tensor $\mathcal{K}$ are expressed as $\mathcal{K}_f = \text{fft}(\mathcal{K}, [\ ], 3)$ and $\mathcal{K} = \text{ifft}(\mathcal{K}_f, [\ ], 3)$, respectively. The block vectorization operation $\text{bvec}(\mathcal{K})$ concatenates all $\mathbf{K}^l$ matrices into a single matrix in $\mathbb{R}^{I_1 I_3 \times I_2}$, while its inverse, $\text{fold}(\text{bvec}(\mathcal{K}))$, restores the original tensor $\mathcal{K}$. Additionally, block circulant matrix is $\text{bcirc}(\mathcal{K})$, and block diagonal matrix is $\text{bdiag}(\mathcal{K}_f)$ [6].

**Definition 1 (Tensor Product)**: Tensor product between two tensors $\mathcal{K} \in \mathbb{R}^{I_1 \times I_2 \times I_3}$ and $\mathcal{Y} \in \mathbb{R}^{I_2 \times I_4 \times I_3}$ is represented as

$$\mathcal{K} * \mathcal{Y} = \text{fold}(\text{bcirc}(\mathcal{K})\text{bvec}(\mathcal{Y})) \tag{3}$$

**Definition 2 (Tensor Singular Value Decomposition, $t$-SVD)**: $t$-SVD of a tensor $\mathcal{K}$ with dimensions $I_1 \times I_2 \times I_3$ is denoted as

$$\mathcal{K} = \mathcal{U}_\mathcal{K} * \mathcal{A}_\mathcal{K} * \mathcal{V}_\mathcal{K}^\mathsf{T} \tag{4}$$

Where two tensors $\mathcal{U}_\mathcal{K} \in \mathbb{R}^{I_1 \times I_1 \times I_3}$ and $\mathcal{V}_\mathcal{K} \in \mathbb{R}^{I_2 \times I_2 \times I_3}$ are orthogonal, and tensor $\mathcal{S}_\mathcal{K} \in \mathbb{R}^{I_1 \times I_2 \times I_3}$ is $f$-diagonal.

**Definition 3 ($\|\mathcal{K}\|_\circledast$ Tensor Nuclear Norm [41]**: Tensor nuclear norm $\|\mathcal{K}\|_\circledast$ represents the summation of singular values across all frontal slices of $\mathcal{K}_f$, articulated as

$$\|\mathcal{K}\|_\circledast = \sum_{l=1}^{I_3} \|\mathcal{K}_f^l\|_\circledast = \sum_{i=1}^{\min(I_1,I_2)} \sum_{l=1}^{I_3} |\mathcal{A}_f^l(i,i)| \tag{5}$$

where $\mathcal{A}_K^l$ is obtained from Eq. (4).

**Definition 4x (Matrix Schatten $p$-Norm ($p \in [0,1]$)) [58]** The matrix Schatten $p$-norm is defined as

$$\|\mathbf{K}\|_{S_p} = \begin{cases} \|\mathbf{K}\|_{S_1} = \|\mathbf{K}\|_* = \sum_{i=1}^{\min\{I_1,I_2\}} \delta_i(\mathbf{K}), \ p = 1 \\ (\sum_{i=1}^{\min\{I_1,I_2\}} \delta_i^p(\mathbf{K}))^{\frac{1}{p}}, 0 < p < 1 \\ \|\mathbf{K}\|_{S_0} = \text{rank}(\mathbf{K}), p = 0 \end{cases} \tag{6}$$

where $\text{rank}(\cdot)$ denotes the operation that calculates the rank of a matrix. When addressing the minimization problem outlined in Eq. (6), it becomes evident that the Schatten $p$-norm $\|\mathbf{K}\|_{S_p}$ with $0 < p < 1$ provides a more effective approach to approximating the rank compared to the nuclear norm $\|\mathbf{K}\|_*$.

### 3.2 Improved weight tensor Schatten $p$-norm

By substituting the matrix nuclear norm defined in Definition 3 with the matrix Schatten $p$-Norm as defined in Definition 4, we arrive at the subsequent formulation:

**Definition 5 ($t$-SVD-based Tensor Schatten $p$-Norm, $t$-TSN [57])** denotes as

$$\|\mathcal{K}\|_{S_p}^p = \sum_{k=1}^{n_3} \left\| \mathcal{K}_f^k \right\|_{S_p}^p = \sum_{i=1}^{\min(n_1,n_2)} \sum_{k=1}^{n_3} \left| \mathcal{K}_f^k(i,i) \right|^p \tag{7}$$

Inspired by **Definition 4**, employing tensor Schatten $p$-norm with $0 < p < 1$ enables a more effective exploration of the low-rank characteristics compared to $t$-TNN [14]. As outlined in [57], the subsequent derivation is as follows:

$$\min_{\mathcal{K}} \eta \|\mathcal{K}\|_{S_p}^p + \frac{1}{2}\|\mathcal{K} - \mathcal{L}\|_F^2 \tag{8}$$

which is solved via the generalized soft-thresholding algorithm [65]. Generally, smaller eigenvalues typically encapsulate less discriminative information than their larger counterparts [9, 57]. This motivates the development of weighted tensor Schatten $p$-norm as

$$\|\mathcal{K}\|_{\omega,s_p}^p = \sum_{k=1}^{I_3} \omega_j^k * \theta_j^p(\mathbf{K}_f^k) \tag{9}$$

Here, $\theta_j^p(\mathbf{K}_f^k)$ and $\omega_j^k$ represent the $j$-th singular value and its corresponding weight of $\mathbf{K}_f^k$, respectively. We determine the weights by inversely scaling them with respect to the singular values, reasoning that larger singular values should undergo less shrinkage. To achieve this, we update $(\omega_j^k)^{t+1} = \frac{1}{(\theta_j^p(\mathbf{K}_f^k))^t + \epsilon}$ using the previous iteration of $\theta_j^p(\mathbf{K}_f^k)$ during optimization, where $t$ denotes the iteration number and $\epsilon$ represents a small constant. Consequently, Eq. (9) is transformed into

$$\min_{\mathcal{K}} \eta \|\mathcal{K}\|_{\omega,s_p}^p + \frac{1}{2}\|\mathcal{K} - \mathcal{L}\|_F^2$$
$$= \min_{\mathbf{K}_f} \sum_{k=1}^{I_3} \eta * \omega_j^k * \theta_j^p(\mathbf{K}_f^k) + \frac{1}{2}\|\mathbf{K}_f^k - \mathbf{L}_f^k\|_F^2 \tag{10}$$

In contrast to the tensor Schatten $p$-norm, our weight tensor Schatten $p$-norm excels in capturing prior information by applying less shrinkage to larger singular values and dynamically updating their associated weights in each iteration. Although effective, Eq. (10) ignores ranking the diagonal core entries of core tensor, where the low-rank structure of core tensor is not fully extracted. To better exploit the low-rank structures hidden in the core tensor and alleviate the noise influence, Eq. (10) further becomes to

$$\min_{\mathcal{K}} \eta \|\mathcal{K}\|_{\mathbb{I}\omega,s_p}^p + \frac{1}{2}\|\mathcal{K} - \mathcal{L}\|_F^2 \tag{11}$$

We devise a new solver for Eq. (11) by using **Theorem 3**, **Theorem 4**, and **Lemma 1**. Note herein that our IWTSN encompasses both the tensor nuclear norm and the tensor Schatten $p$-norm as special cases.

## 3.3 Adaptive Neighbor Bipartite Graph Learning

Given multi-view data $\{\mathbf{X}^r\}_{r=1}^v \in \mathbb{R}^{d^r \times n}$, we extend the formulation presented in Eq. (2) to incorporate adaptive neighbor bipartite graph learning. This extension enables the learning of the anchor matrix $\mathbf{B}^r \in \mathbb{R}^{l \times m}$ and neighbor bipartite graph $\mathbf{A}^r \in \mathbb{R}^{m \times n}$ within the $l$-dimension space of $r$-th latent view.

$$\min_{\mathbf{B}^r, \mathbf{W}^r, \mathbf{A}^r} \sum_{i=1}^m \sum_{j=1}^n \|\mathbf{b}_i^r - (\mathbf{W}^r)^\top \mathbf{x}_j^r\|_F^2 a_{ij}^r + \alpha \|\mathbf{A}^r\|_F^2$$
$$\text{s.t.}(\mathbf{A}^r)^\top \mathbf{1} = 1, \mathbf{A}^r \geq 0, (\mathbf{W}^r)^\top \mathbf{W}^r = \mathbf{I}_m \tag{12}$$

where $\text{Tr}(\cdot)$ is the trace operation. Applying orthogonal constraints to the projection matrices $\{\mathbf{W}^r\}_{r=1}^v \in \mathbb{R}^{d^r \times l}$ and anchor matrices can improve the discriminative power. $\alpha$ is a control parameter to tune the neighbor number around each sample. That is, adjusting the parameter $\alpha \geq 0$ can modulate the structure of the bipartite graph, as stated in Proposition 1 below:

*Proposition 1: By manipulating $\alpha$, a balance between two extreme bipartite graph configurations can be achieved:*

- *Sparse bipartite graphs: each vertex is connected to only one other vertex.*
- *Complete bipartite graphs: all vertices are interconnected with uniform edge weight $1/m$.*

In summary, Proposition 1 elucidates the continuum between sparsity and completeness in bipartite graph configuration through $\alpha$ manipulation. The proof of Proposition 1 refers to [40].

## 3.4 Objective Function Formulation

Although adaptive neighbor bipartite graph learning of Eq. (12) can protect the local structure of data, Eq. (12) is sensitive to noise [11]. Thus, we seamlessly integrate IWTSN of Eq. (9) and Eq. (12) into a unified framework to enhance robustness of bipartite graph, *i.e.*,

$$\min_{\substack{\mathbf{B}^r, \mathbf{W}^r, \\ \mathbf{A}^r, \mathcal{A}}} \gamma \sum_{r=1}^v \sum_{i=1}^m \sum_{j=1}^n \|\mathbf{b}_i^r - (\mathbf{W}^r)^\top \mathbf{x}_j^r\|_F^2 a_{ij}^r + \alpha \|\mathbf{A}^r\|_F^2 + \|\mathcal{A}\|_{\mathbb{I}\omega, S_p}$$
$$\text{s.t.}(\mathbf{A}^r)^\top \mathbf{1} = 1, \mathbf{A}^r \geq 0, (\mathbf{W}^r)^\top \mathbf{W}^r = \mathbf{I}_m, (\mathbf{B}^r)^\top \mathbf{B}^r = \mathbf{I}_m,$$
$$\mathcal{A} = \Psi(\mathbf{A}^1, \cdots, \mathbf{A}^v) \tag{13}$$

where $\gamma$, $\beta$, and $\alpha$ are the control parameters to balance the contributions of respective terms. $\Psi(\cdot)$ denotes tensor stacking function.

## 3.5 Optimization

In order to tackle Eq. (13), we introduce an auxiliary variable $\mathcal{K}$ to enhance the separability of the equation. Subsequently, we reformulate Eq. (13) as the augmented Lagrangian function outlined below.

$$\min_{\substack{\mathbf{B}^r, \mathbf{W}^r, \\ \mathbf{A}^r, \mathcal{Y}, \mathcal{K}}} \gamma \sum_{r=1}^v \sum_{i=1}^m \sum_{j=1}^n \|\mathbf{b}_i^r - (\mathbf{W}^r)^\top \mathbf{x}_j^r\|_F^2 a_{ij}^r + \alpha \|\mathbf{A}^r\|_F^2$$
$$+ \|\mathcal{K}\|_{\mathbb{I}\omega, S_p} + \frac{\mu}{2} \|\mathcal{A} - \mathcal{K} + \frac{\mathcal{Y}}{\mu}\|_F^2 \tag{14}$$
$$\text{s.t. } \mathbf{A}^r \geq 0, (\mathbf{A}^r)^\top \mathbf{1} = 1, (\mathbf{W}^r)^\top \mathbf{W}^r = \mathbf{I}_m, (\mathbf{B}^r)^\top \mathbf{B}^r = \mathbf{I}_m,$$
$$\mathcal{A} = \Psi(\mathbf{A}^1, \cdots, \mathbf{A}^v), \mathcal{A} = \mathcal{K}$$

The optimization problem described by Eq. (14) can be effectively tackled using an alternating iterative algorithm, which is outlined as follows.

▷ **Step-1: Solving A with B, W, $\mathcal{Y}$ and $\mathcal{K}$ fixed.** Optimizing **A** becomes to

$$\min_{a_{ij}^r \geq 0, (\mathbf{a}^r)^\top \mathbf{1} = 1} \gamma \sum_{r=1}^v \sum_{i=1}^m \sum_{j=1}^n \|\mathbf{b}_i^r - (\mathbf{W}^r)^\top \mathbf{x}_j^r\|_F^2 a_{ij}^r$$
$$+ \alpha \|\mathbf{A}^r\|_F^2 + \frac{\mu}{2} \|\mathbf{A}^r - \mathbf{K}^r + \frac{\mathbf{Y}^r}{\mu}\|_F^2 \tag{15}$$

Let $\|\mathbf{b}i^r - (\mathbf{W}^r)^\top \mathbf{x}j^r\|_F^2 = d_{ij}^r$, by eliminating irrelevant variables for the $r$-th view, Eq. (15) changes to

$$\min_{a_{ij}^r \geq 0, (\mathbf{a}_j^r)^\top \mathbf{1} = 1} \|a_{ij}^r - 2\frac{\frac{1}{2}\mu f_{ij}^r - \frac{1}{2}\gamma d_{ij}^r}{(\alpha + \frac{\mu}{2})}\|_F^2 \tag{16}$$

In this context, $f_{ij}^r$ represents an element of $\mathbf{F}^r$, defined as $\mathbf{F}^r = \mathbf{T}^r - \frac{\mathbf{Y}^r}{\mu}$. Subsequently, the update process for $\mathbf{a}^r$ is transformed into a column-wise operation as

$$\min_{\mathbf{a}_j} \|\mathbf{a}_j - \hat{\mathbf{a}}_j\|_F^2, \text{ s.t. } \forall ij, \mathbf{a}_j \mathbf{1} = 1, a_{ij} \geq 0 \tag{17}$$

where $\hat{\mathbf{a}}_j^r = \frac{\frac{1}{2}\mu \mathbf{f}_j^r - \frac{1}{2}\gamma \mathbf{d}_j^r}{(\alpha + \mu)}$. Each column $\mathbf{a}_j$ could be optimized via the following **Theorem 1**.

**Theorem 1.** *With arbitrary $v$ vectors $\{\hat{\mathbf{a}}_j\}_{j=1}^v$, we obtain the following closed-form solution $\mathbf{a}_j^*$*

$$\mathbf{a}_j^* = \arg \min_{\mathbf{a}_j} \|\mathbf{a}_j - \hat{\mathbf{a}}_j\|_F^2, \text{ s.t. } \mathbf{a}_j^\top \mathbf{1} = 1, \mathbf{a}_j \geq 0 \tag{18}$$

whose proof refers to [42].

▷ **Update-2: Solving W with A, B, $\mathcal{Y}$ and $\mathcal{K}$ fixed.** In this case, **W**-subproblem of Eq. (14) can be written as

$$\max_{\mathbf{W}^r} \text{Tr}((\mathbf{W}^r)^\top \mathbf{E}^r) \text{ s.t.} \mathbf{W}^r (\mathbf{W}^r)^\top = \mathbf{I}_k, \tag{19}$$

where $\mathbf{E}^r = \mathbf{X}^r (\mathbf{A}^r)^\top (\mathbf{B}^r)^\top$. Eq. (19) can be solved via the Singular Value Decomposition (SVD) in **Theorem 2** with complexity $O(v\hat{d}(nm + k^2 + km))$ for each iteration, where $\hat{d} = \sum_{p=1}^v d^r$.

**Theorem 2.** [20] *Letting the SVD of $\mathbf{E} \in \mathbb{R}^{d \times l}$ be $\mathbf{E} = \mathbf{UGV}^\top$, where $\mathbf{U} \in \mathbb{R}^{d \times l}, \mathbf{G} \in \mathbb{R}^{l \times l}$ and $\mathbf{V} \in \mathbb{R}^{l \times l}$, the optimal solution of $\max_{\mathbf{W}^\top \mathbf{W} = \mathbf{I}} \text{Tr}(\mathbf{W}^\top \mathbf{E})$ is $\mathbf{W} = \mathbf{UV}^\top$.*

▷ **Step-3 update B:** Optimizing **B** while keeping the irrelevant variables fixed can be understood as

$$\max_{\mathbf{B}^r} \text{Tr}((\mathbf{B}^r)^\top \mathbf{C}^r) \text{ s.t.}(\mathbf{B}^r)^\top \mathbf{B}^r = \mathbf{I}_m \tag{20}$$

in which $\mathbf{C}^r = \gamma (\mathbf{W}^r)^\top \mathbf{X}^r (\mathbf{A}^r)^\top$. The optimal solution for optimizing $\mathbf{B}^r$ can be efficiently obtained using **Theorem 2**.

▷ **Step-4 update $\mathcal{K}$:** Ignoring the irrelevant items *w.r.t.* $\mathcal{K}$, updating $\mathcal{K}$ subproblem is

$$\min_{\mathcal{K}} \|\mathcal{K}\|_{\mathbb{I}\omega, S_p}^p + \frac{\mu}{2}\|\mathcal{K} - (\mathcal{J} + \frac{\mathcal{Y}}{\mu})\|_F^2 \tag{21}$$

According to [30], Eq. (21) can be solved into two steps as follows: (1) minimizing the core matrix, and (2) minimizing $t$-TSN.
(1) Updating core matrix as

$$\min_{\mathfrak{B}(\mathcal{T})} \|\mathfrak{B}(\mathcal{T})\|_* + \frac{1}{2\lambda}\|\mathcal{F} - (\mathcal{J} + \frac{\mathcal{Y}}{\mu})\|_F^2 \tag{22}$$

where regularization parameter $\lambda = 1/(max(m,v)n)^{\frac{1}{2}}$. And the tensor $\mathcal{T}$ is obtained from $t$-SVD on the temporary variable $\mathcal{F}$, i.e., $\mathcal{F} = \mathcal{U} * \mathcal{T} * \mathcal{V}$.

(2) Updating $\mathcal{K}$ as

$$\min_{\mathcal{K}} \; \|\mathcal{K}\|_{\mathbb{I}\omega,S_p}^p + \frac{\mu}{2}\|\mathcal{K}-\mathcal{L}\|_F^2 \tag{23}$$

With the learned low-rank core matrix $\mathfrak{P}(\mathcal{T})$, we can use $t$-product to reconstruct a tensor as $\mathcal{L} = \mathcal{U} * \mathfrak{P}(\mathcal{T})^{-1} * \mathcal{V}$. The learned $\mathcal{L}$ can further produce a closed-form solution via the following **Theorem 3**.

**Theorem 3.** *Consider $\mathcal{L} \in \mathbb{R}^{I_1 \times I_2 \times I_3}$, with $r = \min(I_1, I_2)$. Let $\mathcal{L}_f = \mathcal{U}_f \mathcal{M}_f \mathcal{V}_f^\mathsf{T}$, then the optimization problem for weight tensor Schatten p-norm can be formulated as*

$$\min_{\mathcal{K}} \eta\|\mathcal{K}\|_{\omega,s_p}^p + \frac{1}{2}\|\mathcal{K}-\mathcal{L}\|_F^2 \tag{24}$$

*and its optimal solution is*

$$\mathcal{K}^* = \texttt{ifft}(\mathcal{U}_f * \mathcal{D}_{\eta,\omega,p}(\mathcal{L}_f) * \mathcal{V}_f^\mathsf{T}) \tag{25}$$

*where $\mathbf{M}_f^k = \texttt{diag}(\delta(\mathbf{M}_f^k))$ and $\mathcal{D}\eta,\omega,p(\mathbf{L}_f^k) = \texttt{diag}(\theta(\mathbf{L}_f^k))$ are the k-th frontal slices of $\mathcal{M}_f$ and $\mathcal{D}\eta,\omega,p(\mathcal{L}_f)$ in the Fourier domain with respect to $\theta(\mathbf{L}_f^k) = \texttt{GST}(\delta(\mathbf{M}_f^k),\eta*\omega^k,p)$.*

**Proof:** Eq. (24) becomes to

$$\min_{\mathcal{K}_f} \sum_{l=1}^{n_3}(\sum_{j=1}^{r} \eta * \omega_j^k * \theta_j^r(\mathbf{K}_f^k)) + \frac{1}{2}\|\mathbf{K}_f^k-\mathbf{L}_f^k\|_F^2 \tag{26}$$

in which $\theta_j(\mathbf{K}_f^k)$ denotes the $j$-th singular value of $\mathbf{K}_f^k$, and its corresponding weight is $\omega_j^k = \frac{1}{\theta_j^r(\mathbf{K}_f^k)+\varepsilon}$. Initially, each weight $\omega_j^k$ is set as $\omega_j^k = \frac{1}{\delta_j(\mathbf{M}_f^k)+\varepsilon}$ since $\theta_j^r(\mathbf{K}_f^k)$ is unavailable in the first iteration, and updated based on the previous iteration of $\theta_j^r(\mathbf{K}_f^k)$.

Eq. (26) can be solved separately for different $k$ as

$$\min_{\mathbf{K}_f^k} \sum_{j=1}^{r} \eta * \omega_j^k * \theta_j^r(\mathbf{K}_f^k) + \frac{1}{2}\|\mathbf{K}_f^k-\mathbf{L}_f^k\|_F^2 \tag{27}$$

Solvers are derived using the following **Theorem 4** and **Lemma 1**.

**Theorem 4.** *Consider the singular value decomposition (SVD) of matrix $\mathbf{T} \in \mathbb{R}^{I_1 \times I_2}$ as $\mathbf{T} = \mathbf{U}_A * \mathbf{D}_A * \mathbf{V}_A^\mathsf{T}$, where $\eta > 0, r = \min(I_1, I_2)$, and $0 \le \omega_1 \le \omega_2 \le \ldots \le \omega_r$. The global optimal solution for the following weighted Schatten p-norm minimization problem, adapted from [58], is as follows:*

$$\min_{\mathbf{K}} \eta\|\mathbf{K}\|_{\omega,s_p}^p + \frac{1}{2}\|\mathbf{K}-\mathbf{T}\|_F^2 \tag{28}$$

*As shown in [59], the optimal solution of Eq. (28) is given by*

$$\mathbf{K}^* = \mathbf{U}_A \mathbf{D}_{\eta,\omega,p}(\mathbf{T})\mathbf{V}_A^\mathsf{T} \tag{29}$$

*where $\mathbf{D}_A = \texttt{diag}(\delta)$, $\mathbf{D}\eta,\omega,p(\mathbf{T}) = \texttt{diag}(\theta)$. The vector $\delta = \delta_j(\mathbf{T})j = 1^r$ represents the singular values of $\mathbf{T}$, each of which can be obtained using **Lemma 1** [65].*

**Lemma 1.** *Consider the k-th subproblem of Eq. (28), expressed as*

$$\min_{\theta(\mathbf{K}_f^k) \ge 0} f(\theta_j(\mathbf{K}_f^k)) = \frac{1}{2}(\theta_j(\mathbf{K}_f^k)-\delta_j(\mathbf{L}_f^k))^2 + \eta\omega_j\theta_j(\mathbf{K}_f^k)^p \tag{30}$$

*Within $\omega$ and $p$, soft-thresholding function $\eta p^{GST}(\omega_j)$ is defined as*

$$\eta_p^{GST}(\omega_j) = (2\omega_j(1-p))^{\frac{1}{2-p}} + \omega_j p(2\omega_j(1-p))^{\frac{p-1}{2-p}} \tag{31}$$

*The minimum $Sv^{GST}(\delta_j,\omega_j)$ of Eq. (31) is determined by*

$$T_p^{GST}(\delta_j,\omega_j) = \begin{cases} 0, \; \delta_j < \eta_p^{GST}(\omega_j) \\ \texttt{sgn}(\delta_j)S_p^{GST}(\delta_j,\omega_j), \delta_j \ge \eta_p^{GST}(\omega_j) \end{cases} \tag{32}$$

*in which $S_p^{GST}(\delta_j,\omega_j)$ satisfies*

$$S_p^{GST}(\delta_j,\omega_j) - \delta_j + \omega_j p\left(S_p^{GST}(\delta_j,\omega_j)\right)^{p-1} = 0 \tag{33}$$

*Arranging $\omega$ ($0 \le \omega_1 \le \omega_2 \le \ldots \le \omega_r$) in non-ascending order and $\delta$ ($\delta_1 \ge \delta_2 \ge \ldots \ge \delta_r \ge 0$) in non-descending order aids in determining a global minimizer $\theta$ ($\theta_1 \ge \theta_2 \ge \ldots \ge \theta_r$) using von Neumann's trace inequality, where $r = \min(I_1, I_2)$.*

Updating ADMM variables are written as

$$\begin{aligned} \mathcal{Y} &= \mathcal{Y} + \mu(\mathcal{J}-\mathcal{K}) \\ \mu &= \min(\rho\mu, \mu_{max}) \end{aligned} \tag{34}$$

In the optimization process, we set $\mu = 1e^{-4}$ and $\mu_{max} = 10^{10}$, with a computational complexity of $O(n)$. Algorithm 1 delineates the entire optimization procedure of Eq. (14), wherein convergence is assessed by evaluating the objective value $obj^t$ after the $t$-th iteration.

---

**Algorithm 1** IWTSN-FMGC

---

**Input:** Multi-view data $\{X^r\}_{r=1}^v$, cluster number $c$, latent space dimension $k$, and parameters $\alpha, \gamma$.
    Initialize $\mathbf{Q}^r = \mathbf{I}_k$, and the others matrices as $\mathbf{0}$.
1: **repeat**
2:     Update $\mathbf{A}, \mathbf{W}, \mathbf{B}$, and $\mathcal{K}$ via Eq (15), Eq. (19), Eq. (20), and Eq. (21), respectively;
3:     Update ADMM variables via Eq. (34);
4: **until** Satisfy convergence.
5: Perform SVD on $\hat{\mathbf{A}} = \sum_{r=1}^v \mathbf{A}^r/v$.
**Output:** Clustering metrics.

---

**Space Complexity.** Primary space consumption is to store $\mathbf{W}^r \in \mathbb{R}^{d^r \times l}$, $\mathbf{X}^r \in \mathbb{R}^{d^r \times n}$, $\mathbf{B}^r \in \mathbb{R}^{l \times m}$, $\mathbf{A}^r \in \mathbb{R}^{n \times m}$, $\mathcal{K} \in \mathbb{R}^{m \times n \times v}$, and $\mathcal{A} \in \mathbb{R}^{m \times n \times v}$. Total space complexity of IWTSN-FMGC amounts to $(n+l)m + 3mnv + \hat{d}(n+m)$, which is linear to $n$.

**Table 1: Datasets for experiment evaluation.**

| Dataset | Sample ($n$) | Class ($k$) | View ($v$) | Dimensionality ($d$) |
|---|---|---|---|---|
| YTF-50 | 126054 | 50 | 4 | 944/576/512/640 |
| YTF-20 | 63896 | 20 | 4 | 944/576/512/640 |
| SUNRGBD | 10335 | 45 | 2 | 4096/4096 |
| ORL | 400 | 40 | 3 | 4096 /3304 / 6750 |
| MSRCV1 | 210 | 5 | 6 | 1302 / 48 / 512/100/256/210 |
| Yale | 165 | 15 | 3 | 4096 /3304 / 6750 |
| Synthetic | 100 | 2 | 2 | 2/2 |

**Time Complexity.** Algorithm 1 incurs the following time complexity per iteration: the bipartite graph learning in **Step-1** costs $O(v\hat{d}nm)$; the matrix multiplication and SVD operation of **Step-2** and **Step-3** involve $O(v\hat{d}(nm+l^2+lm))$ and $O(v(n\hat{d}l + m^2l + nml))$, respectively. Improved weighted Schatten $p$-norm optimization

**Table 2: Average Fscore, Purity, NMI, and ACC comparison with 11 SOTA methods on the seven datasets. The bold and blue represent the best and the second-best results, respectively. '-' denotes out of CPU memory or storage memory.**

| Datasets | AMGL[34] | FMR[17] | PMSC[12] | LMVSC[13] | SMVSC[43] | SFMC[22] | FMCNOF[60] | FPMVS[51] | SDAFG[31] | MVBGC[16] | TBGL[57] | Our |
|---|---|---|---|---|---|---|---|---|---|---|---|---|
| ACC | | | | | | | | | | | | |
| YTF-50 | - | - | - | 68.32 ± 2.45 | 69.65 ± 2.46 | - | 21.66 ± 0.00 | 64.24 ± 2.97 | 62.44 ± 0.00 | 68.48 ± 3.12 | 72.31 ± 0.01 | 75.59 ± 0.01 |
| YTF-20 | - | - | - | 67.26 ± 3.53 | 67.13 ± 4.20 | - | 38.61 ± 0.00 | 63.08 ± 3.79 | 61.88 ± 0.00 | 72.24 ± 2.65 | 73.66 ± 0.01 | 77.54 ± 0.02 |
| SUNRGBD | 9.81 ± 0.37 | - | - | 17.87 ± 0.39 | 23.34 ± 0.38 | 11.02 ± 0.00 | 19.67 ± 0.00 | 23.26 ± 0.50 | 16.85 ± 0.00 | 21.13 ± 2.32 | 20.41 ± 0.03 | 24.20 ± 0.01 |
| ORL | 71.15 ± 2.81 | 65.79 ± 3.26 | 63.47 ± 3.09 | 65.65 ± 2.93 | 65.16 ± 1.10 | 50.00 ± 0.00 | 27.50 ± 0.00 | 66.75 ± 0.00 | 73.50 ± 0.00 | 64.53 ± 3.55 | 80.21 ± 0.02 | 95.75 ± 0.01 |
| MSRCV1 | 76.44 ± 6.30 | 77.48 ± 6.40 | 47.45 ± 4.23 | 83.73 ± 7.20 | 70.51 ± 4.98 | 60.48 ± 0.00 | 47.14 ± 0.00 | 71.95 ± 5.36 | 70.95 ± 0.00 | 86.19 ± 7.59 | 90.96 ± 0.00 | 93.10 ± 0.00 |
| Yale | 64.52 ± 4.27 | 68.81 ± 5.95 | 58.80 ± 4.43 | 61.47 ± 3.47 | 66.06 ± 0.00 | 47.27 ± 0.00 | 33.94 ± 0.00 | 67.27 ± 0.00 | 65.45 ± 0.00 | 72.12 ± 0.00 | 88.36 ± 0.04 | 91.81 ± 0.02 |
| NMI | | | | | | | | | | | | |
| YTF-50 | - | - | - | 82.43 ± 0.78 | 83.63 ± 0.85 | - | 43.03 ± 0.00 | 82.08 ± 1.07 | 77.18 ± 0.00 | 83.80 ± 0.96 | 84.68 ± 0.02 | 85.99 ± 0.00 |
| YTF-20 | - | - | - | 76.78 ± 1.34 | 78.36 ± 2.39 | - | 45.45 ± 0.00 | 74.30 ± 1.95 | 73.18 ± 0.00 | 76.59 ± 3.04 | 77.69 ± 0.02 | 79.88 ± 0.02 |
| SUNRGBD | 18.46 ± 0.66 | - | - | 24.50 ± 0.37 | 22.71 ± 0.41 | 2.30 ± 0.00 | 15.66 ± 0.00 | 22.84 ± 0.82 | 11.37 ± 0.00 | 23.82 ± 2.60 | 30.68 ± 0.01 | 37.81 ± 0.00 |
| ORL | 87.64 ± 1.07 | 81.20 ± 1.38 | 80.93 ± 1.39 | 83.35 ± 1.13 | 84.85 ± 0.29 | 81.58 ± 0.00 | 49.23 ± 0.00 | 86.26 ± 0.00 | 88.80 ± 0.00 | 77.41 ± 0.54 | 90.63 ± 0.02 | 99.01 ± 0.00 |
| MSRCV1 | 77.65 ± 3.23 | 69.48 ± 3.31 | 34.29 ± 2.81 | 78.93 ± 4.60 | 62.01 ± 2.61 | 60.23 ± 0.00 | 38.42 ± 0.00 | 65.69 ± 3.27 | 76.23 ± 0.00 | 65.69 ± 3.27 | 85.36 ± 0.01 | 87.32 ± 0.00 |
| Yale | 67.73 ± 1.86 | 74.72 ± 3.38 | 63.74 ± 2.98 | 65.43 ± 1.92 | 69.83 ± 0.00 | 54.27 ± 0.00 | 39.50 ± 0.00 | 71.06 ± 0.00 | 69.20 ± 0.00 | 73.11 ± 0.00 | 86.93 ± 0.01 | 89.98 ± 0.02 |
| Purity | | | | | | | | | | | | |
| YTF-50 | - | - | - | 73.21 ± 2.18 | 72.72 ± 2.61 | - | 22.83 ± 0.00 | 66.84 ± 3.02 | 67.83 ± 0.00 | 75.04 ± 2.33 | 78.91 ± 0.02 | 80.62 ± 0.00 |
| YTF-20 | - | - | - | 73.40 ± 2.75 | 72.40 ± 3.96 | - | 40.34 ± 0.00 | 64.92 ± 3.83 | 68.31 ± 0.00 | 77.14 ± 3.72 | 76.04 ± 0.03 | 80.55 ± 0.02 |
| SUNRGBD | 10.74 ± 0.37 | - | - | 37.42 ± 0.53 | 32.64 ± 0.65 | 11.47 ± 0.00 | 25.15 ± 0.00 | 32.77 ± 1.15 | 17.65 ± 0.00 | 32.69 ± 2.53 | 41.90 ± 0.01 | 49.53 ± 0.00 |
| ORL | 76.47 ± 2.02 | 69.10 ± 2.70 | 67.21 ± 2.73 | 69.18 ± 2.19 | 72.17 ± 1.08 | 79.25 ± 0.00 | 28.25 ± 0.00 | 73.75 ± 0.00 | 79.00 ± 0.00 | 58.97 ± 1.39 | 86.87 ± 0.01 | 96.87 ± 0.01 |
| MSRCV1 | 80.45 ± 4.29 | 79.01 ± 4.16 | 49.91 ± 3.78 | 85.25 ± 5.56 | 71.51 ± 4.02 | 62.86 ± 0.00 | 50.48 ± 0.00 | 72.33 ± 5.01 | 70.95 ± 0.00 | 72.33 ± 5.01 | 88.32 ± 0.02 | 93.10 ± 0.00 |
| Yale | 66.64 ± 3.14 | 70.08 ± 5.39 | 60.47 ± 3.92 | 62.40 ± 3.27 | 66.06 ± 0.00 | 48.48 ± 0.00 | 35.15 ± 0.00 | 67.27 ± 0.00 | 66.06 ± 0.00 | 72.12 ± 0.00 | 89.12 ± 0.00 | 91.81 ± 0.02 |
| F-score | | | | | | | | | | | | |
| YTF-50 | - | - | - | 62.49 ± 2.45 | 63.52 ± 2.56 | - | 15.67 ± 0.00 | 56.89 ± 3.18 | 24.29 ± 0.00 | 61.83 ± 3.35 | 63.88 ± 0.02 | 66.87 ± 0.01 |
| YTF-20 | - | - | - | 62.43 ± 2.91 | 61.68 ± 5.99 | - | 25.84 ± 0.00 | 57.81 ± 4.00 | 42.07 ± 0.00 | 64.38 ± 3.11 | 66.05 ± 0.02 | 70.33 ± 0.02 |
| SUNRGBD | 6.46 ± 0.22 | - | - | 11.41 ± 0.23 | 14.99 ± 0.14 | 12.17 ± 0.00 | 14.08 ± 0.00 | 15.23 ± 0.39 | 13.80 ± 0.00 | 14.96 ± 2.05 | 15.98 ± 0.01 | 18.34 ± 0.00 |
| ORL | 53.73 ± 6.36 | 54.00 ± 3.15 | 52.57 ± 3.06 | 56.52 ± 3.38 | 55.98 ± 0.76 | 32.35 ± 0.00 | 13.80 ± 0.00 | 58.73 ± 0.00 | 47.51 ± 0.00 | 42.87 ± 1.47 | 78.14 ± 0.01 | 95.61 ± 0.01 |
| MSRCV1 | 70.28 ± 4.42 | 66.76 ± 4.50 | 34.05 ± 2.34 | 77.43 ± 6.43 | 59.31 ± 2.82 | 52.43 ± 0.00 | 33.85 ± 0.00 | 61.55 ± 3.54 | 63.58 ± 0.00 | 61.55 ± 3.54 | 83.02 ± 0.01 | 86.66 ± 0.00 |
| Yale | 41.47 ± 3.53 | 56.30 ± 4.88 | 43.23 ± 4.15 | 46.42 ± 2.74 | 52.60 ± 0.00 | 31.28 ± 0.00 | 17.97 ± 0.00 | 54.19 ± 0.00 | 45.91 ± 0.00 | 58.12 ± 0.00 | 80.76 ± 0.02 | 83.52 ± 0.04 |

of **Step-4** is $O(v^2nm + vnm\log(n))$; ADMM variables need $O(v)$. Upon completion, the final $\hat{A}$ requires $O(nm^2)$ for SVD and subsequent $k$-means. Overall, the total time complexity approximates to $O(t(2v\hat{d}nm + vnm\log(n) + nm^2))$. Analogous to space complexity, the time complexity of Algorithm 1 also exhibits linear to sample number. Both space complexity and time complexity are linear to $n$, which enables handling large-scale datasets with $100,000 \leq n$.

## 4 EXPERIMENTS

### 4.1 Experimental Setting

**Benchmark Datasets.** We conducted experiments on six datasets (YTF-50, YTF-20, SUNRGB-D, ORL, MSRCv1, and Yale) to evaluate our approach. The implementations of all competing methods were obtained from their respective public repositories or directly from the authors. Our experiments for shallow methods were carried out on a computing platform consisting of a 32GB RAM and Intel Core i7 CPU, using Matlab 2021b on a 2021 Mac mini.

**Compared Algorithms.** We employed 11 State-Of-The-Art (SOTA) competitors, including AMGL[34], FMR [17], PMSC [12], BMVC [64], LMVSC [13], SMVSC [43], SFMC [22], FMCNOF [60], FPMVS [51], SDAFG [31], UDBGL [8], FastMICE [10], MVBGC [16], and TBGL [57] to assess the effectiveness and efficiency of our IWTSN-FMGC. These competitors were selected based on their performance in four common metrics [13, 50, 54] cited in recent literature.

**Experimental Results.** Table 2 detailedly reports the six evaluated multi-view benchmark datasets by conducting comparative experiments with 11 multi-view clustering methods. From Table 2, we could observe the following:

- Our method consistently surpasses all competitors across all assessed metrics and datasets, underscoring its superiority over existing SOTA multi-view clustering techniques.
- AMGL, FMR, and PMSC leverage a widely adopted self-representation framework for spectral clustering by using full connection self-representation graph learning. However, as evidenced in Table 2, our IWTSN-FMGC consistently outperforms these methods in terms of clustering performance. Meanwhile, IWTSN-FMGC also has very competitive time and space complexity (Linear to sample number).
- Dynamic anchor learning based methods (such as FastMICE, MVBGC, and our IWTSN-FMGC) generally achieve superior clustering performance compared to approaches like LMVSC, SFMC, FMCNOF, SDAFG, and UDBGL, which maintain fixed anchors during optimization. Notably, our IWTSN-FMGC consistently outperforms recently proposed methods like FastMICE and MVBGC, due to its ability to dynamically explore local manifold and high-order information.
- Despite TBGL also leveraging the tensor Schatten $p$-norm to exploit the high-order low-rank structure of bipartite graphs, our IWTSN-FMGC significantly outperforms it across multiple metrics in Table 2. This superiority could be attributed to the dynamic neighbor bipartite graph learning technique of IWTSN-FMGC, which captures local manifold structure more effectively than the fixed bipartite graph method of TBGL. Furthermore, IWTSN-FMGC innovatively develops an enhanced weighted tensor Schatten $p$-norm to better approximate the target rank of the learned bipartite graph tensor, thereby enhancing its innovation and effectiveness.

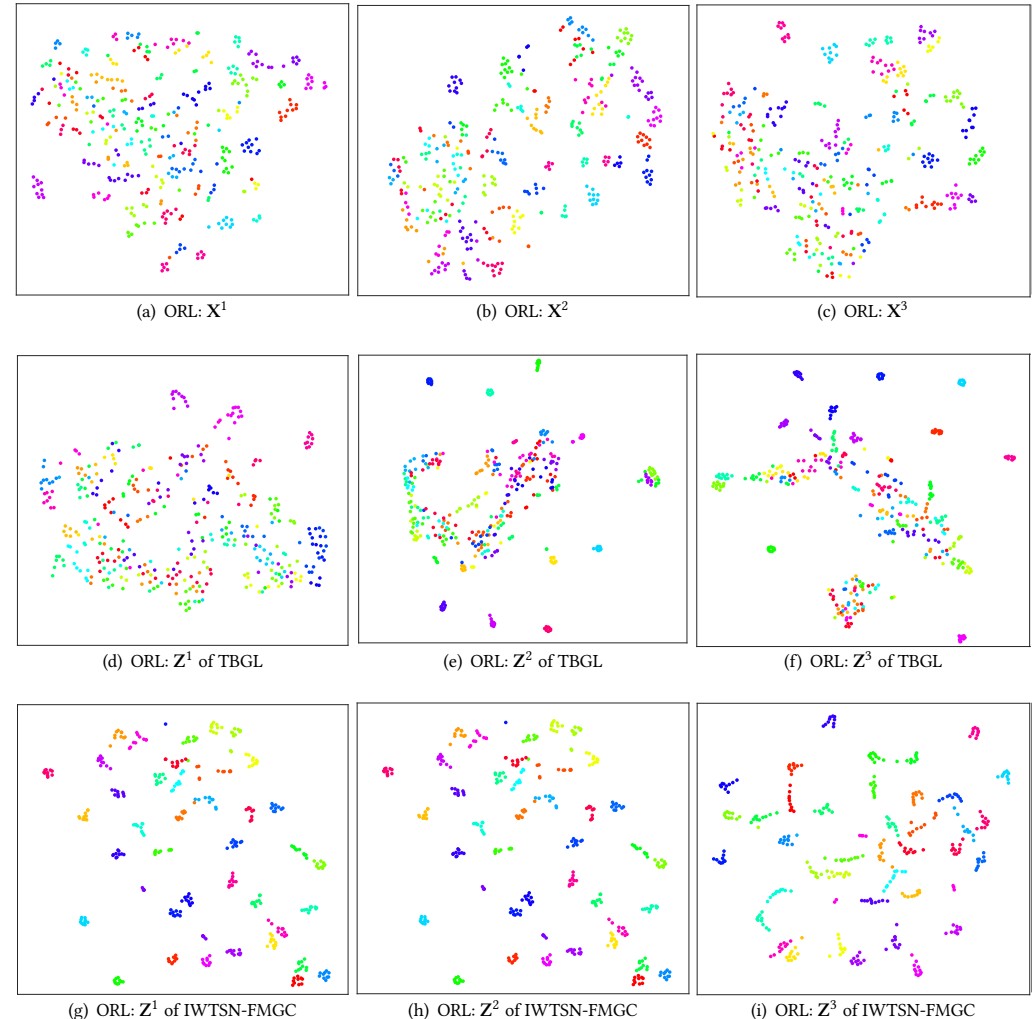

(a) ORL: $\mathbf{X}^1$  (b) ORL: $\mathbf{X}^2$  (c) ORL: $\mathbf{X}^3$

(d) ORL: $\mathbf{Z}^1$ of TBGL  (e) ORL: $\mathbf{Z}^2$ of TBGL  (f) ORL: $\mathbf{Z}^3$ of TBGL

(g) ORL: $\mathbf{Z}^1$ of IWTSN-FMGC  (h) ORL: $\mathbf{Z}^2$ of IWTSN-FMGC  (i) ORL: $\mathbf{Z}^3$ of IWTSN-FMGC

**Figure 1: The original $\{X\}_{r=1}^{v}$, the bipartite graphs $\{Z\}_{r=1}^{v}$ of second best competitor TBGL, and $\{Z\}_{r=1}^{v}$ of our IWTSN-FMGC on ORL dataset are all visualized in Figure 1, where different cluster assignments represent the different colors.**

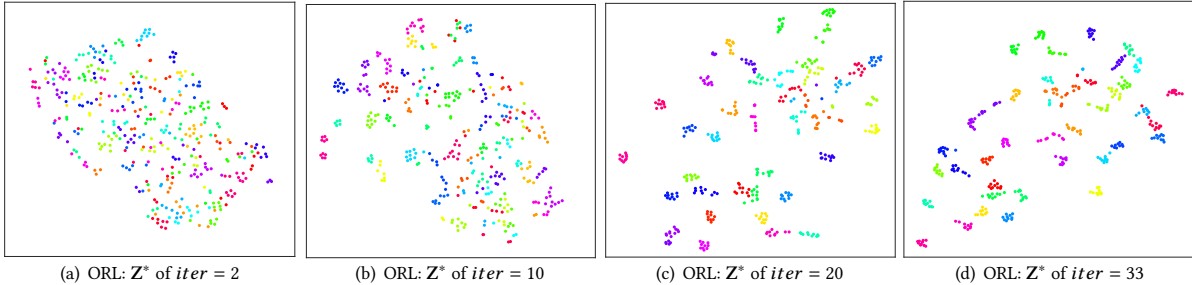

(a) ORL: $\mathbf{Z}^*$ of $iter = 2$  (b) ORL: $\mathbf{Z}^*$ of $iter = 10$  (c) ORL: $\mathbf{Z}^*$ of $iter = 20$  (d) ORL: $\mathbf{Z}^*$ of $iter = 33$

**Figure 2: Visualization of consensus bipartite graph $A^*$ of our IWTSN-FMGC on ORL dataset with iteration increasing, where different cluster assignments represent the different colors.**

## 4.2 Visualization Analysis

Figure 1 (a)-(c) visualize the original $\{\mathbf{X}^r\}_{r=1}^{v}$, while (d)-(i) depict the bipartite graphs $\{\mathbf{A}^r\}_{r=1}^{v}$ generated by TBGL and our IWTSN-FMGC on the ORL dataset, respectively. TBGL and our IWTSN-FMGC exhibit improved cluster structures compared to the original

data. However, TBGL shows scattered structure distributions, leading to incorrect cluster partitions. In contrast, the visualizations (g)-(i) of our IWTSN-FMGC display fewer incorrect cluster assignments than those of TBGL, consistent with the clustering results

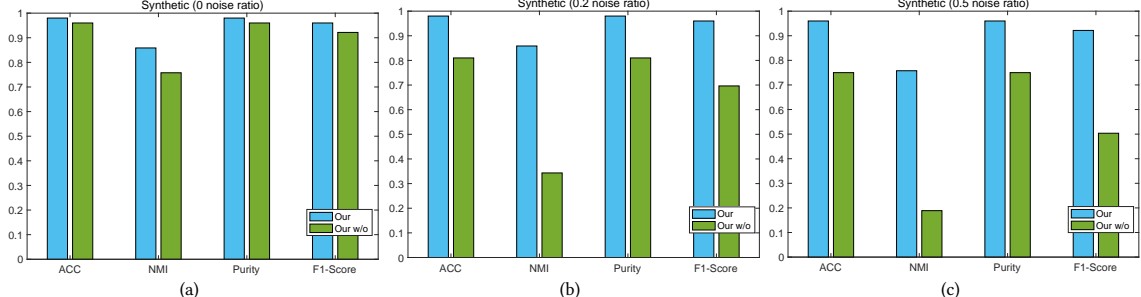

Figure 3: Ablation analysis of our IWTSN on the Synthetic dataset *w.r.t.* different noise ratios.

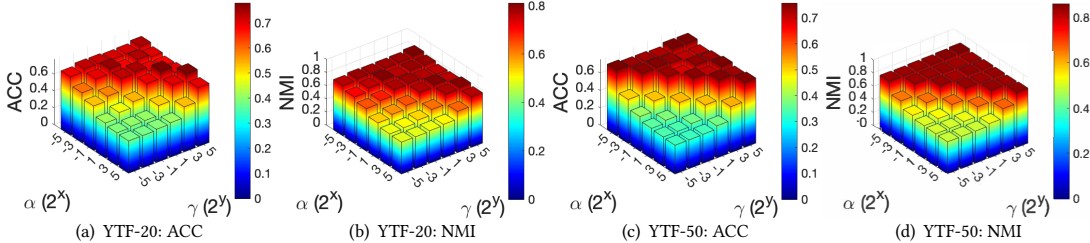

(a) YTF-20: ACC     (b) YTF-20: NMI     (c) YTF-50: ACC     (d) YTF-50: NMI

Figure 4: Parameter analysis on the two datasets.

in Table 2. Additionally, Figure 2 illustrates the evolving clustering structures with increasing iterations, indicating progressively clearer patterns. These results affirm the effectiveness and superiority of our proposed method in learning reliable bipartite graphs.

## 4.3 Ablation Studies

To demonstrate significance, effectiveness, and robustness of tensor Schatten $p$-norm, we degrade our IWTSN $\|\mathcal{A}\|_{\mathbb{I}\omega,s_p}^p$ to $\|\mathcal{A}\|_{s_p}^p$, simplifying as our w/o. By performing experiments on different noise ratios on the Synthetic dataset in Fig. 3, we observe that: 1) our method outperforms our w/o on all evaluated metrics in Fig. 3 (a), indicating the significance and validity of the proposed $\|\mathcal{A}\|_{\mathbb{I}\omega,s_p}^p$; 2) Fig. 3 (b) and (c) demonstrate that our $\|\mathcal{A}\|_{\mathbb{I}\omega,s_p}^p$ outperforms $\|\mathcal{A}\|_{s_p}^p$ by a large margin, verifying the robustness of our method.

## 4.4 Parameter and Convergence

Algorithm 1 incorporates two parameters, $\alpha$ and $\gamma$, which control the influence of neighbor bipartite graph learning and the improvement of weight tensor Schatten $p$-norm $\|\cdot\|_{\mathbb{I}\omega,S_p}$, respectively. These parameters are varied within the range $2^{[-5:2:5]}$, with $m = 2c$ and $l = 2c$ fixed. Specifically, if $d_{min} \leq 2c$, then $l = d_{min}$, where $d_{min}$ represents the minimum dimension across all views. Fig. 4 illustrates the sensitivity analysis of our Algorithm 1 using two large-scale datasets (YTF-50 and YTF-20). The results demonstrate that our IWTSN-FMGC is minimally affected by variations in $\alpha$ and $\gamma$. Moreover, the losses depicted in Fig. 5 consistently decrease to a stable value with increasing iterations on the YTF-20 dataset, confirming robust convergence of our IWTSN-FMGC.

## 5 CONCLUSION

This paper proposes a novel IWTSN to better capture the low-rank property hidden local bipartite graphs. Then, we employ the developed improved weighted tensor Schatten $p$-norm to perform fast local multi-view clustering by integrating the designed adaptive neighbor bipartite graph learning. Meanwhile, a well-designed solution is also provided to solve improved weighted tensor Schatten $p$-norm. Comprehensive experiments and analysis have proved the superiority and validity of our method.

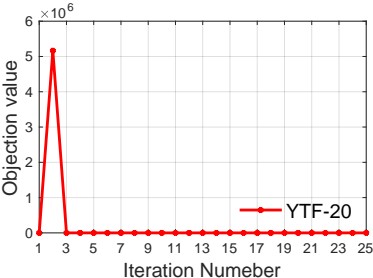

Figure 5: The losses on the YTF-20 dataset.

## ACKNOWLEDGMENTS

This work was supported by the National Natural Science Foundation of China (Grant Nos. 62372235 and 62176055), the China Postdoctoral Science Foundation (Grant No. 2024M750425), the China Scholarship Council (Grant No. 202306840101), the Sichuan Science and Technology Program (Grant Nos. 24ZDZX0007 and 2023NSFSC0506), the Key Lab of Film and TV Media Technology of Zhejiang Province (Grant No. 2020E10015), and the Project of SongShan Laboratory (Grant No. YYJC012022015).

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
