# OpenReview forum: "Improved Weighted Tensor Schatten 𝑝-Norm for Fast Multi-view Graph Clustering"
_acmmm.org/ACMMM/2024/Conference — MM2024 Poster_

### Official Review · Reviewer_Bjoy · 2024-05-20

**Rating:** 5
**Confidence:** 3

**Summary:**

This paper propose a novel Improved Weighted Tensor Schatten 𝑝-Norm for Fast Multi-view Graph Clustering (IWTSN-FMGC). IWTSN-FMGC first develops a novel Improved Weighted Tensor Schatten 𝑝-Norm (IWTSN) with an elegant solver to better rank the core tensor and take advantage of the prior information from singular values, enhancing robustness and validity compared to the widely-used tensor Schatten 𝑝-norm. Then, IWTSN-FMGC seamlessly couples IWTSN and adaptive neighbor bipartite graph learning for fast multi-view graph clustering.

**Strengths:**

Innovation: IWTSN-FMGC presents a new Improved Weighted Tensor Schatten 𝑝-Norm (IWTSN), offering increased robustness and validity over the commonly used tensor Schatten 𝑝-norm. Meanwhile, an elegant solver is developed to optimize the proposed IWTSN.
Experiments: Comprehensive experiments and analysis have proved the superiority and validity of IWTSN-FMGC compared to SOTA competitors.
Scalability: Integrating adaptive neighbor bipartite graph learning with IWTSN synergistically enhances graph quality and robustness, yielding more refined and accurate local manifold structures. Here, IWTSN also has very good scalability to the other graph learning methods.
Code: Code is available to aid reproducibility.

**Limitations:**

1.How to enhance robustness by developing IWTSN?
2.What is the prior information from singular values?
3.What is the difference between adaptive neighbor bipartite graph learning and anchor-based subspace learning.

**Suitability:**

3

---

### Official Review · Reviewer_Ydk1 · 2024-05-21

**Rating:** 3
**Confidence:** 4

**Summary:**

The paper introduces a new approach named Improved Weighted Tensor Schatten 𝑝-Norm for Fast Multi-view Graph Clustering (IWTSN-FMGC). This method aims to address limitations observed in current tensor Schatten 𝑝-norm methods by proposing an enhanced weighted tensor Schatten 𝑝-norm. This enhancement involves dynamically ranking the core tensor and automatically shrinking singular values. As a result, the improved method is expected to better utilize low-rank structures and prior information, potentially improving robustness compared to existing tensor Schatten 𝑝-norm methods. Additionally, the paper proposes an adaptive neighbor bipartite graph learning strategy, offering increased flexibility and expressiveness in encoding local manifold structure information compared to conventional anchor selection approaches. The effectiveness and superiority of the proposed method are demonstrated through extensive experiments conducted on various benchmark datasets.

**Strengths:**

1. The proposed method demonstrates a linear relationship with the number of samples, enabling it to accommodate large-scale datasets.

2. In the experimental section, the proposed method exhibits superior performance compared to several state-of-the-art methods.

**Limitations:**

1. The motivation section of the manuscript's introduction lacks clarity and coherence. While the authors propose an "Improved Weighted Tensor Schatten 𝑝-norm," they fail to provide an overview of any existing works based on Tensor Schatten 𝑝-norm in the introduction. It is worth noting that there are numerous works in this field currently. Additionally, the claim that existing tensor Schatten 𝑝-norm methods cannot fully exploit the low-rank property in the core tensor space lacks substantiation. It would be beneficial for the authors to elaborate on why existing methods fail to exploit this property fully and provide examples of their performance characteristics in this regard.

2. The authors claim that the proposed method is fast, but the experimental section lacks comparisons of runtime to substantiate this assertion.

3. The writing throughout the manuscript could benefit from further refinement.

**Suitability:**

2

---

### Official Review · Reviewer_CkPT · 2024-05-22

**Rating:** 5
**Confidence:** 4

**Summary:**

This paper proposes a multi-view clustering method called Improved Weighted Tensor Schatten 𝑝-Norm for Fast Multi-view Graph Clustering (IWTSN-FMGC). IWTSN-FMGC integrates adaptive neighbor bipartite graph learning and low-rank tensor learning into a unified framework, which can handle large-scale datasets. Additionally, an improved weighted tensor Schatten 𝑝-norm is developed to dynamically rank core tensor and automatically shrink singular values. Extensive experiments are conducted to validate the effectiveness of the proposed IWTSN-FMGC.

**Strengths:**

1. This paper is well-organized, and easy to follow.
2. This paper proposes a new tensor norm with a solver to achieve the greater robustness and effectiveness than the existing tensor norm.
3. This paper proposes a novel large-scale multi-view clustering method by employing the proposed tensor norm.
4. Comprehensive experiments show the superiority of the proposed method over existing SOTA methods.

**Limitations:**

1. The adaptive neighbor graph learning could encode local manifold structure information by selecting the proper neighbors for samples. But for neighbor bipartite graph in this paper, how to select the neighbors.
2. Wether the prior information to enhance the robustness refer to the singular values of core tensor?
3. There are some typos in the paper, such as the suboptimal results under the Purity metric of the SUNRGBD dataset in Table 1 is not marked.

**Suitability:**

2

---

### Official Review · Reviewer_Bwo4 · 2024-05-23

**Rating:** 5
**Confidence:** 4

**Summary:**

This study introduces an innovative Fast Multi-view Graph Clustering model, IWTSN-FMGC. First, an improved weighted tensor Schatten 𝑝-norm (IWTSN) is developed to enhance robustness in adaptive neighbor bipartite graph learning, which is sensitive to noise. Concretely, IWTSN has the potential to more effectively leverage low-rank structures and prior information by dynamically ranking core tensor and automatically shrinking singular values. Then, IWTSN could improve the anchor quality and refine the local manifold structure information in local graph learning process

**Strengths:**

- Despite a wealth of theories and proofs, the manuscript is skillfully written, with concise explanations. The precise detailing ensures concepts are communicated clearly.
-Ranking the core tensor and automatically shrinking singular values for local bipartite graphs are skillful to improve their robustness.
-Comprehensive experiments on real-world and synthetic datasets provide robust empirical evidence, confirming the superior effectiveness of the proposed IWTSN-FMGC.

**Limitations:**

-It is necessary to make a further analysis about time cost comparison on large-scale datasets, such as YTF-50 and YTF-20.
-The paper contains extensive theoretical analysis and proofs. The authors should carefully check the formulas of this manuscript. Such as “E” in Theorem 2 while be $E^r$ in line 437.
-In line 421, superscripts should be consistently formatted.
-I'm confused that why applying less shrinkage to larger singular values for improving the robustness?

**Suitability:**

3

---

### Meta-Review · Area_Chair_jDEG · 2024-07-04

**Recommendation:** Accept (Poster)
**Confidence:** 4

**Metareview:**

The paper received four reviews by confident/very confident reviewers, all of them quite positive, with  final recommendations accept, accept, borderline accept and weak accept.

The first  two reviewers find their doubts solved by the authors in the rebuttal phase. Also the third reviewer is satisfied by the authors’ replies, and indeed increases the final recommendation (from borderline reject to borderline accept). Interestingly, reviewers’ assessment about suitability to the conference ranges between moderately suitable to definitely suitable, but this doesn’t seem to be correlate to the final recommendation.
Taking into account all the mostly positive opinion of the reviewers I recommend acceptance for oral presentation